# Using Machine Learning to Fight Child Acute Malnutrition and Predict Weight Gain During Outpatient Treatment with a Simplified Combined Protocol

**DOI:** 10.3390/nu16234213

**Published:** 2024-12-06

**Authors:** Luis Javier Sánchez-Martínez, Pilar Charle-Cuéllar, Abdoul Aziz Gado, Nassirou Ousmane, Candela Lucía Hernández, Noemí López-Ejeda

**Affiliations:** 1Unit of Physical Anthropology, Department of Biodiversity, Ecology and Evolution, Faculty of Biological Sciences, Complutense University of Madrid, 28040 Madrid, Spain; clhernan@ucm.es (C.L.H.); noemilop@ucm.es (N.L.-E.); 2Action Against Hunger, 28002 Madrid, Spain; pcharle@accioncontraelhambre.org; 3Action Against Hunger, Niamey 11491, Niger; agado@ne.acfspain.org; 4Nutrition Direction, Ministry of Health, Niamey BP 623, Niger; naous002@gmail.com; 5EPINUT Research Group, Faculty of Medicine, Complutense University of Madrid, 28040 Madrid, Spain

**Keywords:** wasting, undernutrition, ENSEMBLE, emergency contexts, Niger

## Abstract

Background/Objectives: Child acute malnutrition is a global public health problem, affecting 45 million children under 5 years of age. The World Health Organization recommends monitoring weight gain weekly as an indicator of the correct treatment. However, simplified protocols that do not record the weight and base diagnosis and follow-up in arm circumference at discharge are being tested in emergency settings. The present study aims to use machine learning techniques to predict weight gain based on the socio-economic characteristics at admission for the children treated under a simplified protocol in the Diffa region of Niger. Methods: The sample consists of 535 children aged 6–59 months receiving outpatient treatment for acute malnutrition, for whom information on 51 socio-economic variables was collected. First, the Variable Selection Using Random Forest (VSURF) algorithm was used to select the variables associated with weight gain. Subsequently, the dataset was partitioned into training/testing, and an ensemble model was adjusted using five algorithms for prediction, which were combined using a Random Forest meta-algorithm. Afterward, Receiver Operating Characteristic (ROC) curves were used to identify the optimal cut-off point for predicting the group of individuals most vulnerable to developing low weight gain. Results: The critical variables that influence weight gain are water, hygiene and sanitation, the caregiver’s employment–socio-economic level and access to treatment. The final ensemble prediction model achieved a better fit (R^2^ = 0.55) with respect to the individual algorithms (R^2^ = 0.14–0.27). An optimal cut-off point was identified to establish low weight gain, with an Area Under the Curve (AUC) of 0.777 at a value of <6.5 g/kg/day. The ensemble model achieved a success rate of 84% (78/93) at the identification of individuals below <6.5 g/kg/day in the test set. Conclusions: The results highlight the importance of adapting the cut-off points for weight gain to each context, as well as the practical usefulness that these techniques can have in optimizing and adapting to the treatment in humanitarian settings.

## 1. Introduction

Child acute malnutrition remains a major public health problem worldwide, and, currently, about 45 million children between 6 and 59 months suffer from it [1]. Malnutrition health effects range from increased comorbidities due to an increased risk of infectious diseases [2] to an increase in mortality [3], but also have long-term effects on the development and growth pattern of an individual [4]. Its complex etiology results from the multiple interrelationships between the factors that are directly related to nutrition, such as food insecurity, feeding practices, inadequate hygiene and sanitation, infectious diseases, etc., as well as other associated variables that include seasonality, access to health services, socio-economic level and gender inequality [5,6].

In recent years, efforts have been made to bring the treatment of acute malnutrition closer to the most affected communities by increasing the treatment provision sites through health posts run by lay Community Health Workers [7]. To optimize resources and facilitate the management of the disease, simplifications of the standard protocols are beginning to be tested, involving changes in the admission, follow-up, and discharge criteria or changes in the nutritional products and their dosage, among others [8]. One of the proposed adaptations is to change the anthropometric criteria used for diagnosis, follow-up and discharge, using only the Mid-Upper Arm Circumference (MUAC) and eliminating the use of weight and height. The MUAC cut-off points that determine admission to treatment and recovery are clearly established since the criteria for the standard protocols are maintained (<115 mm for diagnosed acute cases, 115–125 mm for moderate cases and >125 mm for recovery) [9]. However, the minimum weekly MUAC gain that would allow for identifying non-responders requiring inpatient treatment has yet to be established, so the weight gain in g/kg/day is still used.

The existing World Health Organization (WHO) guidelines recommend treating child acute malnutrition with Ready-to-Use Therapeutic Food (RUTF) for severe acute malnutrition (SAM) and Ready-to-Use Supplementary Food (RUSF) or other specially formulated foods for moderate acute malnutrition (MAM), and using the weight gain to monitor the treatment progression weekly. For outpatient treatment, international standards establish a gain below 5 g/kg/day for 2–3 weeks as the threshold for considering inpatient treatment [9]. This criterion has changed from previous guidelines, which had established it at 8 g/kg/day for severe cases [10]. A systematic review by Das et al. [11], analyzing the effect of alternative approaches for managing acute malnutrition on weight gain, indicated that using RUTF and RUSF could increase the gain compared to alternative dietary approaches, such as energy-dense home-prepared food. The authors also highlighted that the use of prophylactic antibiotics produced a significant increase in weight gain of +0.74 [0.40–1.08] g/kg/day.

Previous studies that have attempted to model the pattern of weight gain during treatment in community settings are scarce in the literature. Simple linear regression models have been applied mainly using anthropometric and clinical variables. Maust et al. [12] found a goodness of fit (R^2^) of 0.25 for their model. Recently, an R^2^ of 0.19 was found in a sample of children with MAM [13], representing quite low adjustment measures, possibly due to the complexity of the contexts in which malnutrition occurs. Among machine learning techniques, ensemble methods have demonstrated in different areas of study that the combination of different predictors allows for a more precise and robust fit to the data, as shown in ([14,15,16], among others). These methods adjust better to the complex realities, since combining different models achieves different predictions, and each one works better on a part of the data due to the specific characteristics of the model or its sensitivity to biases [17].

The objective of the present study is to determine which variables influence weight gain in the treatment of acute malnutrition in children managed under a simplified combined protocol, in order to subsequently adjust an ensemble model capable of the following: (i) predicting at the time of diagnosis whether a child belongs to a risk group for low gain and (ii) optimizing their treatment by personalizing care.

## 2. Materials and Methods

### 2.1. Study Design and Assessment

The present study is a secondary analysis derived from a non-randomized controlled trial testing the effectiveness of using a simplified combined protocol for the decentralized treatment of child acute malnutrition (6–59 months) carried out in the region of Diffa in southeastern Niger [13,18]. The sample consisted of children with uncomplicated acute malnutrition treated as outpatients at the primary health center in the N’Guigmi area and their three related health posts by health staff and Community Health Workers between December 2020 and April 2021 under the ComPAS simplified combined protocol [19]. Regarding the admission criterion for treatment, a child was classified as having SAM if they presented mild edema (+) or if their MUAC was <115 mm. A child was classified as having MAM if their MUAC was between 115 and 125 mm. Treatment was provided with a fixed dose of RUTF of 1 sachet/day for MAM (500 kcal/day, 12.8 g of protein, 30.3 g of lipids, and 45 g of carbohydrates) and 2 sachets/day for SAM (1000 kcal/day, 25.6 g of protein, 60.6 g of lipids, and 90 g of carbohydrates). The discharge criterion was an MUAC > 125 mm at two consecutive visits. During admission to treatment, a series of clinically relevant variables were also collected for each child, including their sex, age, vaccination status, and current comorbidities (fever, vomiting, diarrhea, malaria, or acute respiratory infection).

Diffa is a mainly rural region particularly affected by chronic armed conflicts, which negatively impact food security and livelihood, resulting in thousands of displaced people who have required humanitarian assistance in recent years [20]. To complement the medical information about the treated children, a socio-economic survey was provided to a subsample of caregivers at the treatment facility upon admission, covering 51 variables organized into four dimensions of living conditions: demographics (6), livelihood (25), food security and diversity (14), and access to healthcare (6). There were no inclusion criteria for choosing the respondents. All caregivers who were at the facilities on the same day the supervisors conducted the monthly follow-up visit were invited to respond. The socio-economic information was recorded for a total of 547 children from the 889 enrolled in the main study.

### 2.2. Data Cleaning and Variable Selection

The data management and statistical analysis were conducted using R software v. 4.2.2 [21]. Firstly, the available database was cleaned to obtain a data set with all its information complete, as required by the variable selection phase. This process began with an assessment of the distribution of missing values using Little’s Missing Completely at Random (MCAR) test, which revealed that the missing values were not randomly distributed. A total of 92 individuals with missing data were identified. To handle these values, preserve the sample size, and enhance the generalizability of the findings, imputation methods were applied. This task was performed using the R missForest library, which employs Random Forest models and achieves better results than simpler imputation methods, such as using the mean or mode [22]. Nevertheless, 12 individuals had to be excluded from the analysis because they lacked information on weight gain, the variable of interest being modeled. Therefore, the sample size with socio-economic information resulting from the data preprocessing was 535 children, 241 with SAM and 294 with MAM. This was a sample size that has been shown to be sufficiently robust for applying machine learning models, as further increases to it would result in the models’ efficiency becoming asymptotic, with negligible improvement [23].

The selection of variables was carried out using the Variable Selection Using Random Forest (VSURF) algorithm proposed by Genuer et al. [24], which is structured in three phases: the thresholding step, interpretation step, and prediction step. The algorithm progressively identifies the variables most related to the outcome of interest, which in this study is weight gain, based on the out-of-bag error of the decision trees. Another interesting aspect of the algorithm is that, during the prediction phase, it accounts for whether redundancy has been incorporated into the model, thereby reducing the risk of multicollinearity problems. Figure 1 presents a graphic summary of the process.

### 2.3. Ensemble Model and Cut-Off Point Determination

Ensemble learning aims to build a prediction model by joining the strengths of a collection of simpler base models through stacking [25]. This stacking approach combines two hierarchical levels of prediction: (1) The first level is the classifiers, which are the components of the set and consist of k different models or algorithms that are adjusted on the training data, obtaining k prediction values for each observation. (2) The second level is the classifier or meta-algorithm that combines the k prediction results obtained for each observation by the different components of the previous phase to achieve the final prediction (Figure 1).

Five models based on different statistical bases and available in the R caret package were chosen as the first-level classifiers: Quantile Regression Forest (QRF), K-Nearest Neighbors (KNN), linear model (LM), Principal Components Regression (PCR), and Radial Support Vector Machine (rSVM). The Random Forest algorithm was chosen as the meta-algorithm for the stacking process, and no differential weights were assigned during this process, as no prior information in this area of research suggests that this approach would benefit the overall process or how it would affect the contexts or data for these parameters. The Root Mean Square Deviation was the metric optimized in the ensemble model during the stacking process. The final sample of individuals was randomly divided into a training set (70%), with which the ensemble model was trained, and a test set (30%), on which the model was used to predict the gain of weight and validate it. A repeated cross-validation (cv) was applied to evaluate the prediction value and the reliability of the models in the test set, using the parameters of 10 partitions and 5 repetitions. The performance of the algorithms was evaluated using metrics such as the R^2^ and regression prediction error measures.

To further analyze the classification efficiency of the model, the quantitative values of the weight gain prediction made by the ensemble model were transformed into a dichotomous interpretation of low gain and adequate gain in order to be able to implement the appropriate measures for better care. The method used for this purpose was the preparation of a Receiver Operating Characteristic (ROC) curve, often used to assess a test’s overall diagnostic performance [26]. Different cut-off points were tested in the range of 3–8 g/kg/day, calculating the Area Under the Curve (AUC) associated with each situation to analyze the ensemble model’s diagnostic power depending on the cut-off point to set a low gain.

## 3. Results

The variable selection analysis using the VSURF algorithm, through its progressive phases, identified the variables most related to weight gain during the treatment of acute malnutrition (Table 1). It should be noted that moving from the first selection phase to the subsequent interpretation and prediction phases involves a significant decrease in the number of variables considered, from 27 to 5, respectively. The interpretation phase is the most appropriate for understanding which variables most accurately explain the behavior of weight gain. These variables are the main source of the water supply, the time to get the water supply, the main occupation of the child’s caregiver, the place where people go to satisfy their physiological needs, and other issues regarding access to a health care site. This mixture of variables related to water, sanitation, and hygiene (WASH); employment status; and economic access to a health care site complemented the clinical data obtained at treatment admission to adjust the ensemble model on weight gain using the training data.

Table 2 includes the main parameters for the average error, the fit quality of each algorithm used individually, and the final ensemble model. A substantial improvement can be seen in the parameters produced by the individually adjusted algorithms and those produced by the final ensemble model. A sensitivity analysis demonstrates that improvement is equally important in the model, with the general sample stratified by severity. A reduction in the average error in the adjustment stands out, both in terms of the Mean Absolute Error (MAE) and Root Mean Square Deviation (RMSE), as well as an improvement in the coefficient of determination (R^2^). The Random Forest model is able to measure the importance of each algorithm used once it is adjusted through a variable importance measurement (VIM). In this regard, the VIM was determined based on the “IncNodePurity”, which is a measure of how much the model error increases when a particular variable is randomly permuted or shuffled. Our ensemble model provided the following “IncNodePurity” values: qrf = 5748.955, svmRadial = 4382.534, pcr = 3468.830, knn = 3326.480, and lm = 3292.738.

Once the ensemble model was adjusted, it was used to predict the weight gain in the test data. These predicted values were used to evaluate the diagnostic power of the model using ROC curves, establishing different cut-off points for what would be considered a low weight gain. Figure 2a reflects the evolution of the Area Under the Curve (AUC) for the different cut-off points. The optimal cut-off point is established as 6.5 g/kg/day (marked in red), since it is the highest weight gain at which the AUC value stabilizes and then decreases. Figure 2b shows the ROC curve establishing this new and optimum cut-off point of 6.5 g/kg/day to diagnose a child as having low weight gain with an associated AUC of 0.777 (CI: 0.696; 0.858), which makes the adjusted ensemble model into a good predictor for this cut-off point. For the other performance indicators at this new threshold, sensitivity values of 0.849 and specificity values of 0.627 were achieved. In addition, the optimal weight gain value to predict this group in this specific analysis case was calculated, which turned out to be 5.8. Thus, all the children for whom the ensemble model predicted a gain < 5.8 g/kg/day should be included within the low-gain group.

A characterization of the prediction errors was carried out to better understand these findings and the performance of the ensemble prediction. Figure 3 presents the areas of success and error in the classification, and how far the predicted values are from the observed ones. A good relationship between the predicted and real weight gain values can be identified due to the points’ proximity to the diagonal line, representing a prediction without errors. Furthermore, it reveals a reasonable success rate of 77% (122/158) for the general classification using the ensemble model. However, it is more important to prioritize the classification of individuals who presented low weight gain according to the cut-off point identified as optimal (<6.5 g/kg/day), an objective that is also achieved by the ensemble model, with an accuracy of 84% (78/93).

## 4. Discussion

Malnutrition in children typically develops between 6 and 18 months, a critical period when growth velocity and brain development are especially high. Therefore, in the fight against child malnutrition, it is equally important that interventions are early and effective, to mitigate its negative effects both in the short and long term [27,28].

The weight gain in g/kg/day was established as the key indicator to monitor and evaluate the correct progress of treatment for child acute malnutrition. For outpatient treatment, the different thresholds set are as follows: <5 g/kg/day (poor), 5–10 g/kg/day (moderate), and >10 g/kg/day (good) [29]. Currently, the WHO [9] recommends considering a case as complicated and referring it for hospital treatment if the average weight gain is less than 5 g/kg/day over 2–3 weeks. Nevertheless, studies that have used the standard Community-based Management of Acute Malnutrition (CMAM) protocols have revealed a wide range of values for weight gain. Ashworth [30] reviewed a total of 33 community-based programs for children with SAM, finding that only 33% met the standards of less than 5% mortality and a weight gain greater than 5 g/kg/day. Regarding the studies that have focused on different simplified protocols for acute malnutrition treatment, there is also great heterogeneity; some have reported average values lower than 5 g/kg/day [19,31,32,33], while other studies have reported figures that exceeded this reference value [18,34]. It should be noted that these studies that found gains above 5 g/kg/day also applied a simplification in the RUTF dosage that implies a reduction in the amount used concerning standardized protocols, and still obtained an adequate average weight gain.

This heterogeneity of the average gain values in published studies reflects the complexity of weight gain behavior, which does not follow a linear pattern throughout treatment. The weight gain velocity during treatment shows a quickly decreasing pattern, in which the first weeks represent a high catch-up in weight. From the fifth treatment week onward, the weight gain velocity drops to <2 g/kg/day [35,36]. In addition, the multiple factors that influence these gains must be considered, including the differences between the protocols used in each study, adherence to it, or even external variables that have been shown to be associated with weight gain, such as feeding practices and comorbidities [37].

Likewise, there is much controversy about the optimal weight gain velocity during recovery. Some recent studies have found an association between very rapid weight gain and cardiovascular problems in adulthood [38,39], arguing it happens through potential excess fat deposition during acute malnutrition treatment, related to the high-fat content of RUTF combined with a quick weight gain pattern. However, a randomized controlled trial (RCT) conducted by Kangas et al. [40] does not support these results, as they found, in a group of children treated for SAM, that their fat mass had not caught up by the end of treatment and remained below that of community controls and British reference data. Another study developed by Fabiansen et al. [41] in Burkina Faso found that children with MAM mainly gained fat-free tissue (93.5%) when rehabilitated. Likewise, another RCT by Suri et al. [42] showed a healthy pattern of weight gain, with the majority being fat-free weight (~80%). Therefore, more longitudinal studies are needed to clarify this hypothesized link, using several observational points over time and controlling for the many confounding factors.

Recently, newer approaches are revolutionizing how child acute malnutrition is diagnosed and treated, greatly simplifying protocols. This has opened the door to considering whether the MUAC gain could be an equally valid and appropriate indicator for tracking a patient’s recovery. Although this aspect has barely been explored, some studies have suggested that the values for weight and MUAC are parallel in their evolution during SAM treatment [43,44,45]. Furthermore, changes in the MUAC and weight have been observed to occur similarly and rapidly during episodes of illness occurring during treatment, with no lag effect on the part of the MUAC [46]. One argument in favor of using the MUAC is that this measure is less affected by hydration status than weight, thus giving a more stable and reliable value for monitoring [47]. Another aspect that explains the potential of the MUAC for community treatment is its simplicity, since it requires fewer and cheaper tools, and no reference tables are needed, unlike weight-based follow-up. Likewise, the MUAC has been identified, along with weight-for-age, as the anthropometric index that best predicts mortality risk, above that of weight-for-height [48].

In recent years, there has been a growing amount of research utilizing machine learning techniques to predict various maternal and child health-related conditions, such as nutritional status, anemia, and mortality, with results that surpass those of traditional statistical methods [49,50,51]. Many of them have focused on exploring which algorithm provides the best classification rates or the best prediction errors. However, ensemble approaches have not been sufficiently explored in these areas, despite having been demonstrated in numerous analysis cases that the combination of algorithms provides better results than the exploration and subsequent individual use of them. To our knowledge, the only study apart from the current one that has used these methodologies is that of Khan and Yunus [16], which successfully improved the prediction rates for malnutrition among children under five compared to individual algorithms. Applying machine learning to accurately identify the profiles of cases at higher risk of low weight gain could be especially useful in implementing simplified protocols based only on the MUAC. Without the need to monitor weight, the most vulnerable cases could be identified at the beginning of treatment so that special attention could be paid to their evolution, or so that treatment could be reinforced with other complementary assistance.

To our knowledge, the present study is the first in the literature to empirically highlight the need to adapt the cut-off point used for each specific context, and the relevance that this adaptation may have for predictive power and practical usefulness. The results provided in the present study indicate that among a large group of socio-economic variables, those most closely linked to weight gain are related to the source of water consumption, sanitation and hygiene, the caregiver’s employment–socio-economic level, and access to treatment. Numerous studies have linked factors of these types with the prevalence and severity of acute malnutrition [49,52,53,54]. The results of our study seem to indicate that their effects also last during treatment, affecting the ability of treated children to recover at a greater or lesser velocity. However, these findings are based on outpatient treatment with a simplified combined protocol in emergency settings in rural Niger, and more studies with the same approach are needed in other contexts.

## 5. Conclusions

The present study has identified a group of variables associated with low weight gain during the outpatient simplified treatment of child acute malnutrition in rural areas of Niger, declared as emergency contexts. Ensemble models are an appropriate tool for predicting low weight gain at the time of diagnosis, and, by establishing the optimal cut-off point, they prove to be good classifiers when predicting new data. The results here reveal the benefits of these methodologies for the optimization and personalization of the treatment of malnutrition, thus anticipating its harmful effects on the health of affected children.

## Figures and Tables

**Figure 1 nutrients-16-04213-f001:**
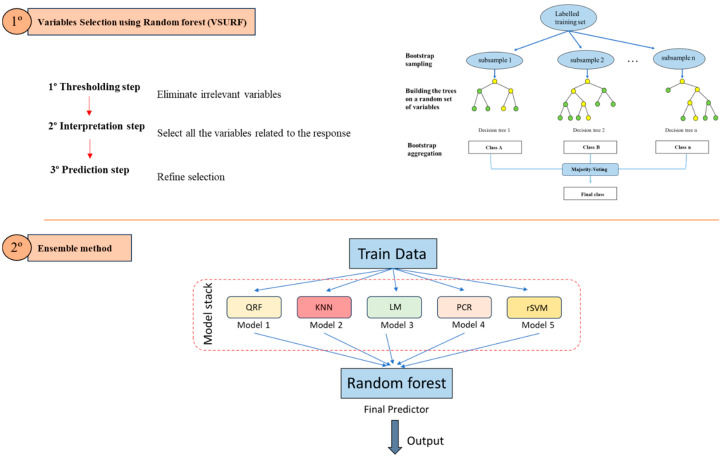
The workflow of the steps followed in selecting the variables and the subsequent adjustment of the ensemble model. QRF: Quantile Regression Forest; KNN: K-Nearest Neighbors; LM: linear model; PCR: Principal Components Regression; rSVM: Radial Support Vector Machine.

**Figure 2 nutrients-16-04213-f002:**
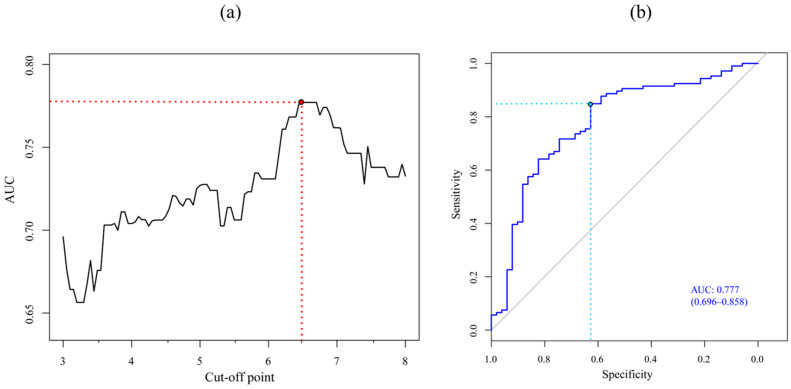
Prediction efficacy of low weight gain group using Receiver Operating Characteristics curve analysis. (**a**) Evolution of AUC modifying cut-off point to diagnose low weight gain. The red dotted line identifies the key cut-off point and AUC values. (**b**) ROC curve establishing cut-off point for low weight gain of 5.5 g/kg/day.

**Figure 3 nutrients-16-04213-f003:**
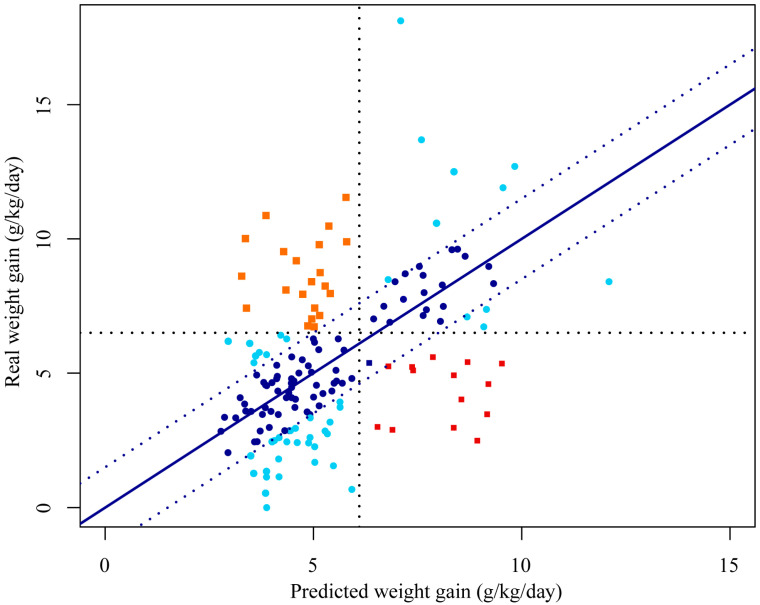
Comparison between the predicted and real values of weight gain. The dotted vertical and horizontal lines indicate the identified cut-off points. Shapes: circles—correct classifications; squares—incorrect classifications. Colors: dark blue—±2 g/kg/day; light blue—>±2 g/kg/day; red—incorrect and overestimated; orange—incorrect and underestimated.

**Table 1 nutrients-16-04213-t001:** The set of variables retained after data cleaning for the analysis of the VSURF algorithm and whether they were selected in each of the phases related to weight gain.

Variable	Possible Responses	VSURF Variable Selected	VSURF Variable of Interpretation	VSURF Variable of Prediction
Caregiver’s relationship to the child.	Grandmother, Mother, Father, Sister, Aunt, Other	No	No	No
Age of caregiver.	numeric	Yes	No	No
Sex of caregiver.	Male, Female	No	No	No
Age of mother when pregnant.	numeric	Yes	No	No
Number of children < five years.	numeric	Yes	No	No
Number of prenatal visits.	None, 1, 2, 3, 4 or more, DKW	No	No	No
Supplements during pregnancy (iron and/or folic acid)?	Yes, No, DKW	No	No	No
Has the child ever been breastfed?	Yes, No, DKW	No	No	No
Is the child currently being breastfed?	Yes, No, DKW	No	No	No
Main source of water supply.	Purchased water, rainwater, surface water, unprotected well, protected well, home tap, community tap, water tank truck, other	Yes	Yes	Yes
Distance to get your water supply?	Less than 100 m, between 100 and 300 m, between 300 and 500 m, more than 500 m, DKW	Yes	No	No
Time to get your water supplies?	Less than 15 min, 15–30 min, 30 min–1 h, 1–2 h, more than 2 h, DKW	Yes	Yes	Yes
Debugging water.	Yes, No, DKW	No	No	No
Where do you go to satisfy your physiological needs?	In the fields/bushes/rivers, latrine, DKW	Yes	Yes	Yes
Main occupation of the head of the household.	Agriculture, house employee, office employee, cattle raising, transport, unskilled manual labor, skilled manual labor, sales and service, other, DKW	Yes	No	No
Main occupation of the child’s caregiver.	Agriculture, house employee, office employee, cattle raising, transport, unskilled manual labor, skilled manual labor, sales and service, other, DKW	Yes	Yes	Yes
How many days of the week does the caregiver perform this job?	Between 1 and 5 days, between 6 and 7 days, DKW	Yes	No	No
How many hours does the child’s caregiver perform this job?	Less than 8 h, more than 8 h, DKW	Yes	No	No
How many people live in the child’s household?	numeric	Yes	No	No
What is the status of your household?	In propriety, for rent, on loan, other, DKW	Yes	No	No
Main construction material for the roof of your household.	Concrete/cement/sheet metal, dung/mud and grass, branches/grass/leaves, other, no roof, DKW	Yes	No	No
Main construction material for the floor of your household.	Concrete/cement/sheet metal, dung/mud and grass, branches/grass/leaves, other, no floor, DKW	No	No	No
Does your household have a separate room for the kitchen?	Yes, No, DKW	No	No	No
Do you have a mattress?	Yes, No, DKW	Yes	No	No
Do you have a mobile phone?	Yes, No, DKW	Yes	No	No
Do you have a refrigerator?	Yes, No, DKW	No	No	No
Do you have a television?	Yes, No, DKW	No	No	No
Do you have a radio?	Yes, No, DKW	Yes	No	No
Do you have a sewing machine?	Yes, No, DKW	No	No	No
Do you have a table?	Yes, No, DKW	No	No	No
Do you have chairs or benches?	Yes, No, DKW	No	No	No
Do you have access to electricity?	Yes, No, DKW	Yes	No	No
Do you have access to livestock?	Yes, No, DKW	No	No	No
Do you have access to arable land?	No, less than 1 ha, between 1 and 5 has, more than 5 has, DKW	Yes	No	No
Household’s main food source.	Purchase on credit, purchase with money, purchase in kind, help from relatives, NGO food assistance, gathering/hunting or fishing, own production, DKW	No	No	No
Enrolled in a food assistance program?	Yes, No, DKW	No	No	No
Enrolled in a cash assistance program?	Yes, No, DKW	No	No	No
Enrolled in a water, hygiene, or sanitation assistance program?	Yes, No, DKW	No	No	No
Enrolled in any assistance program?	Yes, No, DKW	No	No	No
During the year, has the number of meals at home been reduced?	Yes, No, DKW	Yes	No	No
In the last 4 weeks, have you been worried about a lack of food?	No, Rarely (1 or 2 times), Sometimes (3 to 10 times), Often (more than 10 times), DKW	Yes	No	No
In the last 4 weeks, have you reduced your usual portion of food?	No, Rarely (1 or 2 times), Sometimes (3 to 10 times), Often (more than 10 times), DKW	Yes	No	No
In the last 4 weeks, have you reduced the number of usual meals?	No, Rarely (1 or 2 times), Sometimes (3 to 10 times), Often (more than 10 times), DKW	Yes	No	No
In the last 4 weeks, have you gone an entire day without eating?	No, Rarely (1 or 2 times), Sometimes (3 to 10 times), Often (more than 10 times), DKW	Yes	No	No
What do you usually do when your child is sick?	Healer, self-medication, health post/CHW, traditional medicine, nothing, other, DKW	No	No	No
Do you think there are barriers to access to treatment for malnutrition?	Yes, No, DKW	No	No	No
What is your means of transportation to get to the health center?	Car, Motorbike, Bike, Donkey, Charrette, Walking, Other	Yes	No	No
How long does it take to get to the health center?	Less than 1 h, between 1 and 2 h, more than 1 h, DKW	Yes	No	No
Can you make the round trip to the health center in one day?	Yes, No, DKW	No	No	No
Other things you have to pay for when you go to the health center?	Yes, No, DKW	Yes	Yes	Yes
Food Diversity Index.	Poor, Limited, Acceptable	No	No	No

**Table 2 nutrients-16-04213-t002:** Parameters for the quality of the fit of the individual algorithms used and the final ensemble model combining them.

	Model	MAE	RMSE	R^2^
Individual models	QRF	1.99	2.89	0.27
	KNN	2.31	3.11	0.14
	LM	2.23	3.12	0.18
	PCR	2.15	2.93	0.22
	rSVM	2.15	2.95	0.23
Ensemble model	General	1.58	2.26	0.55
	SAM	1.53	2.02	0.61
	MAM	1.58	2.28	0.42

MAE: Mean Absolute Error; RMSE: Root Mean Square Deviation; QRF: Quantile Regression Forest; KNN: K-Nearest Neighbors; LM: linear model; PCR: Principal Components Regression; rSVM: Radial Support Vector Machine; SAM: severe acute malnutrition; MAM: moderate acute malnutrition.

## Data Availability

The original contributions presented in this study are included in the article. Further inquiries can be directed to the corresponding author.

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
