# Peer review of "Using Machine Learning to Fight Child Acute Malnutrition and Predict Weight Gain During Outpatient Treatment with a Simplified Combined Protocol"

_nutrients, 2024, doi:10.3390/nu16234213_

Round 1

Reviewer 1 Report

Comments and Suggestions for Authors

The research aims to forecast weight increase in children experiencing acute malnutrition through machine learning methodologies. It functions under a streamlined methodology that relies on mid-upper arm circumference for diagnosis, follow-up, and discharge, rather than weight. The study, conducted in the Diffa region of Niger, employs various socioeconomic data gathered at admission to predict weight increase results, utilizing an ensemble model of machine learning algorithms to improve predictive accuracy.

1. What methodology was employed to ascertain that the sample size was statistically adequate to guarantee reliable machine learning predictions, particularly in light of the high complexity of the dataset comprising 58 socioeconomic variables?

2. Considering the substantial dropout rate resulting from absent data (decreasing from 547 to 455 children), how could this attrition bias influence the generalizability of the model and its predictions?

3. The manuscript indicates the utilization of an ensemble model that integrates five distinct machine learning techniques. Could you elucidate the methodology employed for assigning weights to various models within the ensemble, as well as the criteria utilized to assess their individual contributions?

4. Could you furnish additional information regarding the performance metrics of each individual algorithm within the ensemble model prior to their integration? This would aid in comprehending the relative efficacy of the employed algorithms.

5. The ideal cut-off point for weight gain was established utilizing ROC curves. Nevertheless, there is an absence of references to sensitivity, specificity, or other performance indicators at this threshold. Could these measurements be supplied to enhance the evaluation of the model's clinical utility?

6. Given the substantial influence of socioeconomic determinants on weight increase prediction, how do you address potential multicollinearity among these predictors, and what effect could this have on the model's predictions?

7. The research is based on data gathered from a particular regional context in Niger. What is your strategy for externally validating the model in diverse geographical and cultural contexts with potentially major socioeconomic disparities?

8. What procedures have been implemented to guarantee the interpretability of machine learning model predictions for healthcare practitioners in clinical settings?

9. A "meta-algorithm" is referenced in the ensemble model. Could you elaborate on the characteristics of this algorithm and its particular function in amalgamating the outputs of the separate models?

10. The shift from employing both weight and MUAC to exclusively utilizing MUAC in malnutrition regimens represents a substantial alteration. How does the model tackle any differences or issues in assessing malnutrition severity based exclusively on MUAC, particularly in borderline instances?

Author Response

First of all, on behalf of the group of authors we would like to thank the reviewer for his/her time and work in agreeing to review our manuscript, as well as for all questions and suggestions.

Comments 1:

What methodology was employed to ascertain that the sample size was statistically adequate to guarantee reliable machine learning predictions, particularly in light of the high complexity of the dataset comprising 58 socioeconomic variables?

Response 1:

As stated in Materials and Methods (line 94), this study is a secondary analysis of data from a previous study that tested the effectiveness of a simplified combined protocol, the main variable considered in the study was treatment recovery. To assess the post-hoc statistical power of the main study, the sample size was calculated using the Fisher’s exact test for comparing two binomial proportions in two independent groups, under a 5% α error probability and a two-tail hypothesis of inequality

(formulas available in the attached document)

On the other hand, a comment on the sample size in machine learning models has been added to the revised manuscript (lines 134-137). This aspect was evaluated in the study by Rajput et al. (2023), which demonstrated through simulations that there exists a sample size n beyond which the efficiency of the models becomes asymptotic, and further improvements are negligible. As shown in Figures of the aforementioned work, a sample size of around 500 individuals, similar to that in our study, already reaches these optimal n values, ensuring the most reliable results in machine learning.

Rajput, D.; Wang, W.J.; Chen, C.C. Evaluation of a decided sample size in machine learning applications. BMC Bioinformatics 2023 24(1), 48. https://doi.org/10.1186/s12859-023-05156-9.

Comments 2:

Considering the substantial dropout rate resulting from absent data (decreasing from 547 to 455 children), how could this attrition bias influence the generalizability of the model and its predictions?

Response 2:

Based on this assessment and the suggestion made by another reviewer to apply an imputation methodology, we have optimized the study's final sample to 535 children (12 had to be necessarily eliminated because they lacked information on weight gain, which is the variable to be modeled). This process was carried out using the R library missForest, which applies Random Forest models. All this information has been added in the new manuscript (lines 124-133).

Therefore, we believe that we have optimized the sample size and potentially enhanced the generalizability of the findings. The analyses were repeated with this expanded sample, yielding nearly identical results in terms of outcomes and interpretation, which have been updated into the revised version of the manuscript.

Comments 3:

The manuscript indicates the utilization of an ensemble model that integrates five distinct machine learning techniques. Could you elucidate the methodology employed for assigning weights to various models within the ensemble, as well as the criteria utilized to assess their individual contributions?

Response 3:

Adjusting the ensemble model, the Random Forest algorithm was selected as the meta-algorithm for the stacking process. No differential weights were assigned during the amalgamation of the models (stacking), as there is no prior information in this research area suggesting this would benefit the ensemble process, nor specifying the contexts or exact data where such an approach would be advantageous. Consequently, the Random Forest meta-algorithm treats the outputs of each independent algorithm equally. Information was added in the new manuscript (lines 159-161).

However, in relation to this, the caretStack function, from the R library ‘caretEnsemble’, used for stacking includes a ‘metric’ argument. This argument specifies the metric to be used for grid search on the stacking model, enabling the optimization of the ensemble model selection based on the lowest Root Mean Square Deviation (RMSE). This last information was incorporated into the revised version of the manuscript (lines 161-163).

The Random Forest model is able to measure the importance of each of the variables used once it is adjusted (in this case they would be algorithms). For our specific example, the calculation of IncNodePurity that is a measure of how much the model error increases when a particular variable is randomly permuted or shuffled was: qrf = 5748.955, svmRadial = 4382.534, pcr = 3468.830, knn = 3326.480, lm = 3292.738. We can observe how the qrf and svmRadial models are the ones that acquire the most importance, the remaining three being very similar. This information was included in the new manuscript (lines 210-215).

Comments 4:

Could you furnish additional information regarding the performance metrics of each individual algorithm within the ensemble model prior to their integration? This would aid in comprehending the relative efficacy of the employed algorithms.

Response 4:

Table 2 presents three metrics evaluating the performance of individual algorithms applied to regression. The R2 value, widely regarded as the primary fit metric for regression models, provides a relative measure (0-1) of each algorithm's prediction efficiency and is easy to interpret. Additionally, the table includes the Mean Absolute Error (MAE) and the Root Mean Square Deviation (RMSE). The RMSE, calculated as the square root of the mean squared error, is easier to interpret since it is expressed in the original units and represents the average prediction error of the models. It has been specified in material and methods that the performance of each algorithm has been evaluated using different metrics (lines 167-168).

In summary, we think that since this is a regression problem, the metrics of the algorithms used are sufficiently covered in Table 2. On the other hand, if these were classification algorithms (which is not the case) other metrics such as Accuracy, sensitivity, specificity, etc. must be provided.

Comments 5:

The ideal cut-off point for weight gain was established utilizing ROC curves. Nevertheless, there is an absence of references to sensitivity, specificity, or other performance indicators at this threshold. Could these measurements be supplied to enhance the evaluation of the model's clinical utility?

Response 5:

The optimal cut-off point for weight gain to achieve an AUC of 0.777 in the ROC curve is 6.5 g/kg/day, which provides a sensitivity of 0.849 and a specificity of 0.627. This cut-off point prioritizes identifying cases (children who are truly wasted), which is clinically advantageous, as a large proportion of children with reduced weight gain according to official standards (<5.5 g/kg/day) will be correctly identified by the model. This information has been included in the revised manuscript (lines 235-236).

Comments 6:

Given the substantial influence of socioeconomic determinants on weight increase prediction, how do you address potential multicollinearity among these predictors, and what effect could this have on the model's predictions?

Response 6:

This aspect is addressed in the initial stage of variable selection using the Variable Selection Using Random Forest (VSURF) algorithm. As mentioned in the Materials and Methods section (lines 138–140), the algorithm consists of three successive phases that progressively refine the selection of variables most associated with the variable of interest, in this case, weight gain. With respect to addressing potential multicollinearity between these predictors, VSURF distinguishes between its interpretation phase and its prediction phase in this regard, as mentioned by Genuer et al. (2015) in the abstract of their article: “The first is a subset of important variables including some redundancy which can be relevant for interpretation, and the second one is a smaller subset corresponding to a model trying to avoid redundancy focusing more closely on the prediction objective…”, the details of this process are specified throughout the paper.

A clarification on this detail of the VSURF algorithm and the advantage it offers in addressing possible multicollinearity between predictors has been introduced in the revised version of the manuscript (lines 142-145).

Genuer, R.; Poggi, J.M.; Tuleau-Malot, C. VSURF: An R Package for Variable Selection Using Random Forests. The R Journal 2015, 7(2), 19-33.

Comments 7:

The research is based on data gathered from a particular regional context in Niger. What is your strategy for externally validating the model in diverse geographical and cultural contexts with potentially major socioeconomic disparities?

Response 7:

The authors recognize the complexity of the etiology and treatment of child acute malnutrition, as well as the significant influence of different contexts. Therefore, as a limitation, we have included this section at the end of the discussion as a concluding remark: “…these findings are based on outpatient treatment with a simplified combined protocol in emergency settings of rural Niger, and more studies with the same approach are needed in other contexts”. In other words, the present study should serve as a working proposal to be applied and adapted to specific contexts, rather than providing universal results applicable to all situations. The findings from the rural context of Niger, where we have identified influences of water consumption, sanitation and hygiene, the caregiver's employment-socioeconomic level and access to treatment, may not necessarily apply to all settings. However, the proposed methodologies for variable selection (VSURF) and subsequent ensemble model adjustment are sufficiently robust to be adapted to new contexts and improve the management of child acute malnutrition.

Comments 8:

What procedures have been implemented to guarantee the interpretability of machine learning model predictions for healthcare practitioners in clinical settings?

Response 8:

As part of the ongoing project conducted by NGO staff, the results will be shared within working groups (clusters) that include ministry staff, researchers from national centers such as CERMES (Centre de Recherche Médicale et Sanitaire), and representatives from other NGOs and United Nations agencies. The content will be tailored to ensure clarity and encourage questions from all participants.

A next step for the future would be to incorporate algorithms into the diagnostic process. This application would allow for the entry of data from a new child diagnosed with acute malnutrition. The app would then predict the expected weight gain, and based on the result, either the standard treatment or a more personalized one could be selected. The NGO collaborating on the project associated with this study has experience working with applications of this type to combat acute child malnutrition.

Comments 9:

A "meta-algorithm" is referenced in the ensemble model. Could you elaborate on the characteristics of this algorithm and its particular function in amalgamating the outputs of the separate models?

Response 9:

In stacking, a meta-algorithm (in this case we use Random forest model) is trained on the predictions of base models. This approach allows the meta-algorithm to learn patterns in model errors and combine them in an optimal way for the regression task. This contrasts with the improvement in the regression metrics of the ensemble model compared to the individual algorithms, as evidenced by both the increase in R2 and the reduction in error (see Table 2).

Random forest models are themselves ensemble algorithms, since they work by adjusting n decision trees through bagging, which are then combined to obtain the final prediction result. The bagging process follows the following steps (Breiman, 2001):

1 – Bootstrap Sample Creation: Multiple training subsamples are generated from the original dataset using the bootstrap method. In each subsample, instances are randomly selected from the dataset with replacement.

2 – Tuning Decision Trees: A base model is trained from each subsample. In the case of Random Forest, these base models are decision trees. Each tree is built using a random subset of variables at each node, which helps diversify the models. At the meta-algorithm level, the variables correspond to the five base algorithms.

3 – Aggregation of Results: For regression problems, the average of the predictions from the n trees is taken. In our case, we specified in the caretStack function that we wanted to minimize the RMSE error.

Breiman, L. Random Forests. Machine Learning 2001, 45 (1), 5–32. https://doi.org/10.1023/A:1010933404324.

Comments 10:

The shift from employing both weight and MUAC to exclusively utilizing MUAC in malnutrition regimens represents a substantial alteration. How does the model tackle any differences or issues in assessing malnutrition severity based exclusively on MUAC, particularly in borderline instances?

Response 10:

In fact, the shift toward using only MUAC in the management of child acute malnutrition is not surprising, and the current research in this direction is becoming stronger. As noted in the introduction (lines 49-62), significant attention and effort have been devoted over the years to simplifying treatment protocols for child acute malnutrition. The validity and importance of MUAC are indisputable because: (i) it is a much simpler tool to use, (ii) it has the highest correlation with mortality, and (iii) its cut-off points are optimized to diagnose the majority of affected individuals. Therefore, the exclusive use of MUAC in the model for assessing malnutrition severity is justified, as it is a well-established tool that offers additional benefits in complex contexts, such as those examined in the present study. This has been addressed in two previous articles demonstrating the effectiveness of this new treatment model in both severe and moderate cases (Charle-Cuellar et al., 2023; Sánchez-Martínez et al., 2023).

Charle-Cuéllar, P.; Lopez-Ejeda, N.; Gado, A.A.; Dougnon, A.O.; Sanoussi, A.; Ousmane, N.; Lazoumar, R.H.; Sánchez-Martínez, L.J.; Touré, F.; Vargas, A.; Guerrero, S. Effectiveness and Coverage of Severe Acute Malnutrition Treatment with a Simplified Protocol in a Humanitarian Context in Diffa, Niger. Nutrients 2023, 15(8), 1975. https://doi.org/10.3390/nu15081975.

Sánchez-Martínez, L.J.; Charle-Cuéllar, P.; Gado, A.A.; Dougnon, A.O.; Sanoussi, A.; Ousmane, N.; Lazoumar, R.H.; Toure, F.; Vargas, A.; Hernández, C.L.; López-Ejeda, N. Impact of a simplified treatment protocol for moderate acute malnutrition with a decentralized treatment approach in emergency settings of Niger. Front. Nutr 2023, 10, 1253545, https://doi.org/10.3389/fnut.2023.1253545.

Reviewer 2 Report

Comments and Suggestions for Authors

The exclusion of 92 incomplete cases from the dataset significantly reduces the sample size and may introduce selection bias, especially if the missing data is not random (e.g., missingness is associated with factors like socioeconomic status or health conditions). We suggest conducting an evaluation of the missing data patterns to assess their randomness. Methods like Little’s MCAR test or comparisons of characteristics between included and excluded cases could help determine if the excluded data might bias the results. Alternatively, consider using imputation methods to handle missing values, which could preserve the sample size and potentially enhance the generalizability of the findings.

Author Response

First of all, on behalf of the group of authors we would like to thank the reviewer for his/her time and work in agreeing to review our manuscript, as well as for all questions and suggestions.

Comments 1:

The exclusion of 92 incomplete cases from the dataset significantly reduces the sample size and may introduce selection bias, especially if the missing data is not random (e.g., missingness is associated with factors like socioeconomic status or health conditions). We suggest conducting an evaluation of the missing data patterns to assess their randomness. Methods like Little’s MCAR test or comparisons of characteristics between included and excluded cases could help determine if the excluded data might bias the results. Alternatively, consider using imputation methods to handle missing values, which could preserve the sample size and potentially enhance the generalizability of the findings.

Response 1: 

We fully understand your concerns and have proceeded as suggested to directly address the problem associated with missing values. First, we applied the Little's Missing Completely at Random (MCAR) Test on the dataset of 547 individuals, obtaining significant results, evidence against MCAR. This indicated the need to apply data imputation methodologies, which were carried out using the R missForest library that employs Random Forest models (detailed in lines 124-133 of the new manuscript). This method was chosen because it achieved better results than simpler imputation methods, such as those using the mean or the mode. With this method, we were able to avoid excluding the 92 individuals with incomplete data. However, 12 of them had to be eliminated because they had missing values (NA) in the target variable to be modelled (weight gain). The results were repeated with this new expanded sample of individuals, reaching almost identical results in terms of results and interpretation, which have been introduced in the new version of the manuscript.

Round 2

Reviewer 1 Report

Comments and Suggestions for Authors

The authors have replied to my comments satisfactorily. 

Reviewer 2 Report

Comments and Suggestions for Authors

I thank the authors for the changes they have made.

They agree  with the requests. 

In my opinion the work can be accepted in this form